# Genome-wide data suggest a revision in management of the Caspian Sea trout *Salmo caspius*

Arezo Najafikhah[1], Mehrshad Zeinalabedini[2]*, Babak Arefnezhad[3], Mohsen Mardi[4], Maryam Nafari[5], Maryam Nasrolahpourmoghadam[6], Omid Jafari[7]*

**1** Department of Biotechnology, Payame Noor University, Tehran, Iran, **2** Department of Genomics, Agricultural Biotechnology Research Institute of Iran (ABRII), Karaj, Iran, **3** OMICS Research Group (Media Teb Gene Co. Ltd.), Tehran, Iran, **4** Seed and Plant Certification and Registration Institute of Iran, Agricultural Research, Education and Extension Organization (AREEO), Karaj, Iran, **5** Iranian Fisheries Organization, Tehran, Iran, **6** Faculty of Natural Resources, Department of Fisheries Sciences, University of Tehran, Karaj, Iran, **7** International Sturgeon Research Institute, Iranian Fisheries Science Research Institute, Agricultural Research, Education and Extension Organization, Rasht, Iran

* mzeinolabedini@abrii.ac.ir (MZ); jaafari.omid@yahoo.com (OJ)

**Data Availability Statement:** The GBS data of Salmo caspius used in this study is available at

## Abstract

*Salmo caspius* Kessler, 1877 is one of the most commercially important species of Salmonidae in the southern basin of the Caspian Sea. The occurrence of its wild populations has undergone sever reduction during the last years. In spite of the yearly restocking activity, still no progress on the recovery of its wild population has been observed. Hence, the present study was done in order to assess the efficiency of the current restocking activity in the southern Caspian basin in term of genetic diversity. DNA extracts of 32 *S. caspius* from four different groups were screened using 62621 genome-wide single nucleotide polymorphisms (SNP). The overal genetic diversity and Fst values were 0.18 and 0.08, respectively. Considering the observed admixture pattern and the positive values for inbreeding coeficient it seems that *S. caspius* suffers from its small effective population size. In order to obtain the maximum performance, alonside with expanding the size of brood stocks, rehabilitation of the habitats and spawning rivers of this nationally endangered species is of great importance.

## Introduction

Salmonidae, is one of the most commercially important fish families with a worldwide distribution. Many of the wild resources of Salmonidae have undergone stock enhancement programs as a consequence of sever reduction in population size [1]. Due to wide geographical distribution and allopatric isolation of brown trout alongside with their morphologic and ecologic variations, natural and/or human-caused hybridization the taxonomy level of *salmo* trouts has been under challenge [2]. Based on the latest reported check list on Iranian fresh water fish species, there are two species of Salmonidae in the southern basin of the Caspian Sea, namely *Salmo trutta* Linnaeus, 1758 (river resident) and *Salmo caspius* Kessler, 1877 (sea-run) which was mostly supported through morphologic approaches [3,4]. Having said that,

Sequence Read Archive (SRA) with accession no PRJNA966795.

**Funding:** We are grateful to Agricultural Research, Education and Extension Organization (AREEO) for providing the required financial and technical resources. MZ was the main recipient of the funding award. The identification number for this funding is: 134-05-0557-008-94006-940029. The funders had no role in study design, data collection and analysis, decision to publish, or preparation of the manuscript.

**Competing interests:** The authors have declared that no competing interests exist.

recently published papers using simultaneous genomics and mtDNA information suggested that all populations of brown trout in the south and south-west of the Caspian Sea should be identified as *S. caspius* [5–7].

There are two ecological forms of *S. caspius* in the southern basin of the Caspian Sea. First, sea-run which migrates to the rivers for natural breeding during spring and fall seasons, however no genetic data supported any differences between fall and spring breeders of *S. caspius*. Secondly, the permanent resident of fresh water drainages which mostly are distributed in basins namely the Caspian Sea, Namak Lake, and Uromia Salt Lake. During the last decades, the capture rate of *S. caspius* throughout the southern basin of the Caspian Sea have had a considerable reduction. There are several factors for this observation among which the overfishing, poaching, drought, rivers' system manipulation, water pollution and dam constructions draw more attention [8,9]. It is presumed that the populations of *S. caspius*, the lake-run form of *Salmo* in the southern Caspian basin, have been more affected by the dam constructions and consequent losing nursery grounds, making it too difficult for natural breeding and recovery of the wild stocks [10]. To compensate this reduction, the Iranian stock management department made a semi-artificial breeding program on this species by capturing the wild mature specimens migrating to the rivers and releasing-back the following fingerlings to the rivers. Historically, the first experience of Caspian salmon artificial breeding backs to the 1967 in Kelardasht breeding center and since then annually tens of thousands of fingerlings are being released to the rivers of the southern basin of the Caspian Sea. Although, based on the observed sea catch it seems that *S. caspius* has not yet been recovered properly and considered endangered at the national level [11]. There are two morphotypes of *S. caspius* based on the seasonal breeding, namely fall and spring breeders between October-November and March-April, respectively, however molecular investigations did not prove the significant differentiation and less is known about the morphology differences between these two forms [12].

The main goal of stock assessment and restocking programs are to identify distinct genetic stocks and keep the standing genetic variations between different populations in fish of the same species [13]. Hence, using molecular information in stock management and aquaculture has been inevitable part of any successful conservation program [14]. The genetic diversity and relationship parameters obtained from genetic data provide a powerful tool to watch-through and maintain the genetic health in the brood stocks and natural populations. In the study conducted on *Salmo trutta* using eight microsatellite loci, the importance of standing genetic diversity in the brood stock against the whirling disease of generated offspring was reported [15]. Different molecular techniques have been used in fish population studies since more than three decades ago, among which SSRs have been the most ideal markers compared to other traditional techniques [16]. Nowadays, because of the impressive advances in sequencing technologies known as Next Generation Sequencing (NGS), researchers are able to have accessibility to the genomic information of both model and nonmodel organisms [17,18]. Genotyping-by sequencing (GBS) is a technique on the basis of genome complexity reduction by restriction enzymes which has been widely applied in fish population studies and aquaculture activities [19,20]. For example, large scale SNP obtained from GBS were successfully used in separation wild and cultured populations of *Cynoglossus semilaevis* [21] and significant QTL affecting resistance to Koi Herpesvirus was detected through a genome-wide association study (GWAS) in Common Carp [22]. The ongoing restocking program of *S. caspius* has been based on random mating of breeders without knowledge on their genetic relationships, and concerning the small effective population size of this species the final output of the restocking can be affected. Hence, the present study was done using GBS-obtained SNP markers to assess the efficiency of the current restocking activity of *Salmo caspius* in Kelardasht restocking center in the southern basin of the Caspian Sea.

## Materials and methods

### Sampling

Before sampling, fish specimens were anesthetized by immersion in 6 mg/l concentration of clove oil (Sigma Aldrich, St. Louis, Missouri, United States,). A total of 31 fin tissues were randomly collected from wild and captivated brood stocks of *S. caspius* in the restocking center at Kelardasht in 2018 (**Fig 1**). The specimens were from four groups hereafter named as NCP (New wild candidate parents; aiming to be used for the ongoing breeding, n = 3); CPLY (Candidate parents used in the last year practice, n = 9); CO (Candidate offspring; offspring generated by the last year breeding, n = 15) and CP (Captured parents; the wild breeding stock of Caspian salmon caught in the past, n = 4). The fin tissue samples were stored in absolute ethanol and transferred to the genetic lab of Systems Biology department at ABRII for the later use in molecular studies.

### DNA extraction

The DNA extraction of the fin tissue samples was done with Qiagen DNeasy Blood and Tissue kit using the provided protocol by the manufacturer (https://www.qiagen.com/us). After the DNA extraction, quality in terms of integrity and pollution and quantity of the extracted DNA samples were assessed by NanoDrop ND-1000 (https://www.thermofisher.com/ca/en/home.html) and 1% agarose gel electrophoresis.

### GBS library preparation and sequencing

The method for GBS library preparation and sequencing is fully described in the paper on common carp (*Cyprinus carpio*) [19]. The paired-end GBS sequencing of the samples was

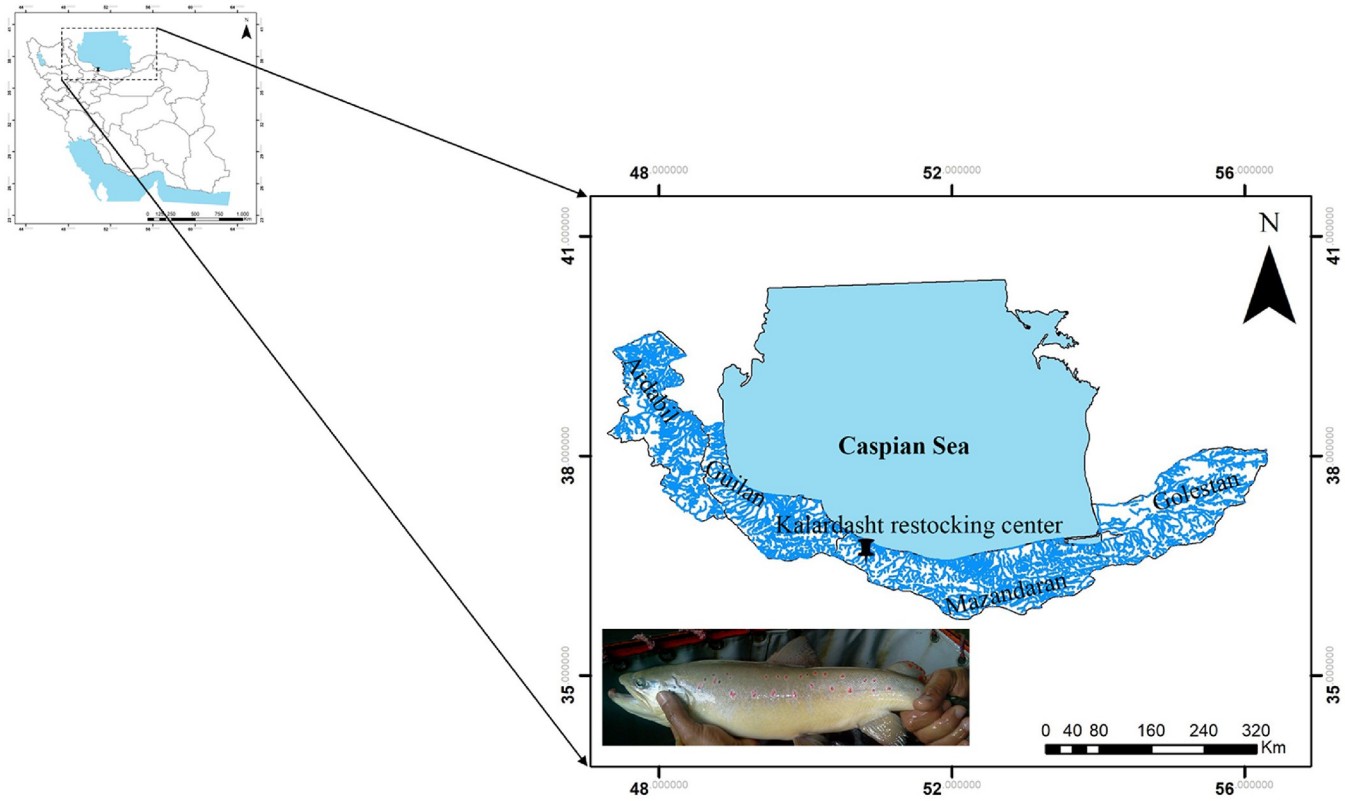

**Fig 1. Restocking center of Caspian salmon at Kalardasht, Iran.**

done at BGI company in China (https://www.bgi.com/global). Briefly, the Apek I enzyme (NEB, USA) was used to digest 100ng of each DNA sample. The specific common and barcode adapters were ligated to the produced DNA fragments and incubated at 22˚C for 1h. In the next step, equal volumes of the DNA products from each sample were pooled and purified using the QIAquick PCR Purification kit (Qiagen). The enrichment of adapter-ligated fragments was done through PCR amplification using PCR Primer Cocktail and PCR Master Mix. PCR products between 180-480bp were selected on an agarose gel. Agilent Technologies 2100 Bioanalyzer and the ABI StepOnePlus Real-Time PCR System were used for quality control assessment of the libraries. At the end, the 150bp PE sequencing of the GBS libraries was done in a lane of Illumina Hiseq 2000 at BGI.

## Bioinformatic analyses

**Quality filtering and SNP calling.**    The FASTQC ver. 0.11.5 was used for quality control of the fastq files (Babraham Bioinformatics, Babraham Institute (www.bioinformatics.babraham.ac.uk/projects/fastqc/). Stacks pipeline version 2 [23] was implemented for the following SNP calling. The process rad tag package with the default parameters was used for raw reads filtering and omitting the low-quality reads. The clean reads were then mapped to the recent genome assembly of *salmo trutta* (NCBI; GCF_901001165.1) using Bowtie2 ver. 2.3.2 with activating the —very-sensitive option [24]. The sorted BAM files were fed into gstacks to build loci from the PE reads, remove the PCR duplicates and call the SNPs. Finally, the Populations module of the Stacks 2 was used to generate a Variant Call Format (VCF) file with the loci available in at least 65% of the animals (-r 0.65), filtering possible sequencing errors, paralogous sequence variants (PSVs) and uninformative polymorphisms. HWE and ld filtering were done through the GBS_SNP_Filter package [25]. Regarding the ld treatment, merely one locus of the loci in linkage disequilibrium (with $r^2 > 0.5$) was remained in the final VCF file. The PGDSpider 2.1.1.5 software was used to convert the generated VCF file into required formats [26].

**Population structure analysis and genetic diversity.**    Three different analytical methods were used for cluster analysis. Firstly, an individual-based Principal Component Analysis (PCA) was done in R [27] package ape [28]. The dispersal of the fish individuals was visualized using the first two components by ggplot2 package [29]. Secondly, a Discriminant Analysis of Principal Components (DAPC) was done on the SNP data aiming to maximize the variance among groups. Thirdly, the STRUCTURE analysis was performed in R using adegenet package [30], considering the lowest Bayesian information criterion (BIC = 256.28) for estimating the number of optimum populations (K = 2).

The Nei-based genetic distances among individuals and different groups were calculated using the poppr package [31]. The fineRADSTRUCTURE was also used to evaluate the level of genetic structuring based on shared coancestry [32,33]. The Fst index between different groups of *S. caspius* was extracted using Genepop 4.0.9 [34]. Observed ($H_o$) and expected heterozygosity ($H_e$) indices were calculated using package GBS_SNP_Filter [25]. To avoid bias due to small sample size, allele frequency differences (AFD) between pairs of groups were calculated [35]. AMOVA was run in Genodive 2.0b23 with 9999 permutations, considering distances based on ploidy independent Infinite Allele Model (Rho) [36]. Before doing AMOVA, a pre-replacing of missing values was performed by randomly selecting values based on overall allele frequencies.

## Results

The final VCF file contained 62621 variant SNPs after different steps of filtering. The proportion of missing data for each individual is available in S1 Table. PC1 and PC2 showed 9.63% of

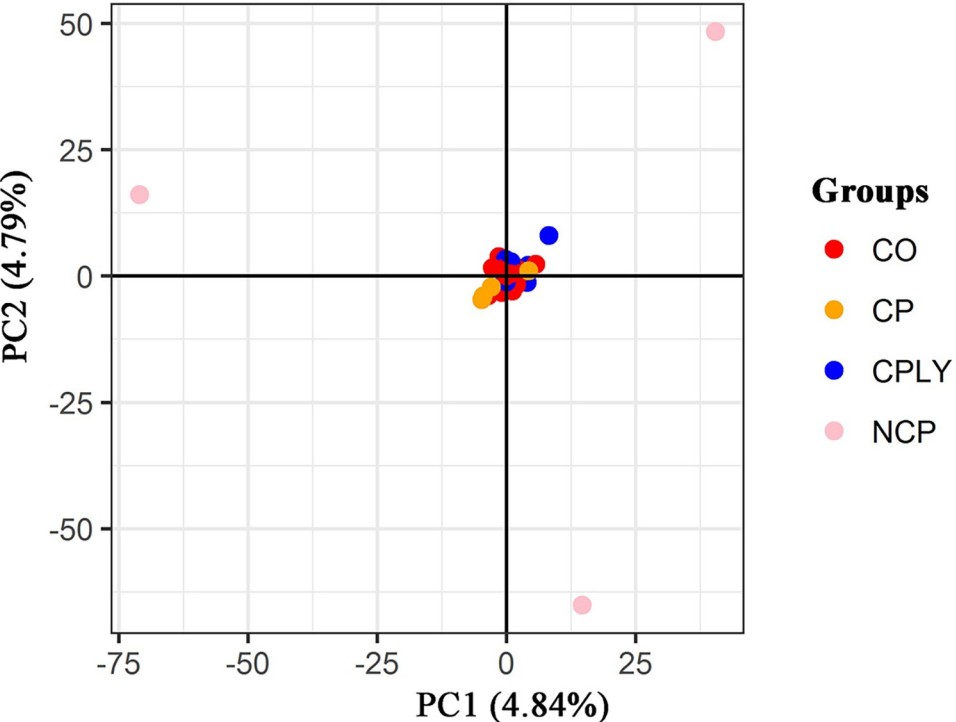

**Fig 2. PCA scatter plot of *S. caspius* brood stocks based on the two first components (CO: Candidate offspring, CP: Captured parents, CPLY: Candidate parents, NCP: New candidate parents).**

the total variance and the PCA results based on the two first components illustrated that the fish individuals of the NCP had a wider dispersal and were far away from the other groups (**Fig 2**). Similar to the PCA, the DAPC scatter plot separeted the NCP group from the other *S. caspius* individuals (**Fig 3**). The genetic background of the investigated fish individuals revealed a high level of admixture structure for the all investigated fish specimens except for the NCP (**Fig 4**).

The observed heterozygosity ($H_o$) as one of the indices of genetic diversity had its maximum value in the NCP group ($H_o = 0.30$), while the CO group had the lowest value with $H_o = 0.12$ (**Table 1**). The maximum number of private alleles were identified in the CPLY group while the NCP showed the less regarding this index. The highest and lowest rate of inbreeding values were recognized in the CO and NCP groups with Fis values of 0.72 and 0.24, respectively. Details information of genetic divesity paramters obtained from SNP data on *S. caspius* are available in **Table 1**. Pairwise Fst values for the between group comparisons ranged from 0.04 to 0.16 for CPLY-CO and NCP-CP comparisons, respectively (**Table 2**). Moreover, AFD range was from 0.03 to 0.187 for the NCP-CP copmparison (**Table 2**). The AMOVA assigned 95% of variance to within-pop variations while among-pop variations accounted for 5% (p = 0.001). Nei-based genetic diversity showed the maximum genetic diversity between NCP and CP groups (**Fig 5**), however, we observed some indications of within-population differentiation using fineRADSTRUCTURE analysis to identify fine-scale genetic clustering. Accordingly, the fineSTRUCTURE classification based on the common shared genome among individualas (coancestry history) detected three main and seven sub-groups of *S. caspious* (**Fig 6**).

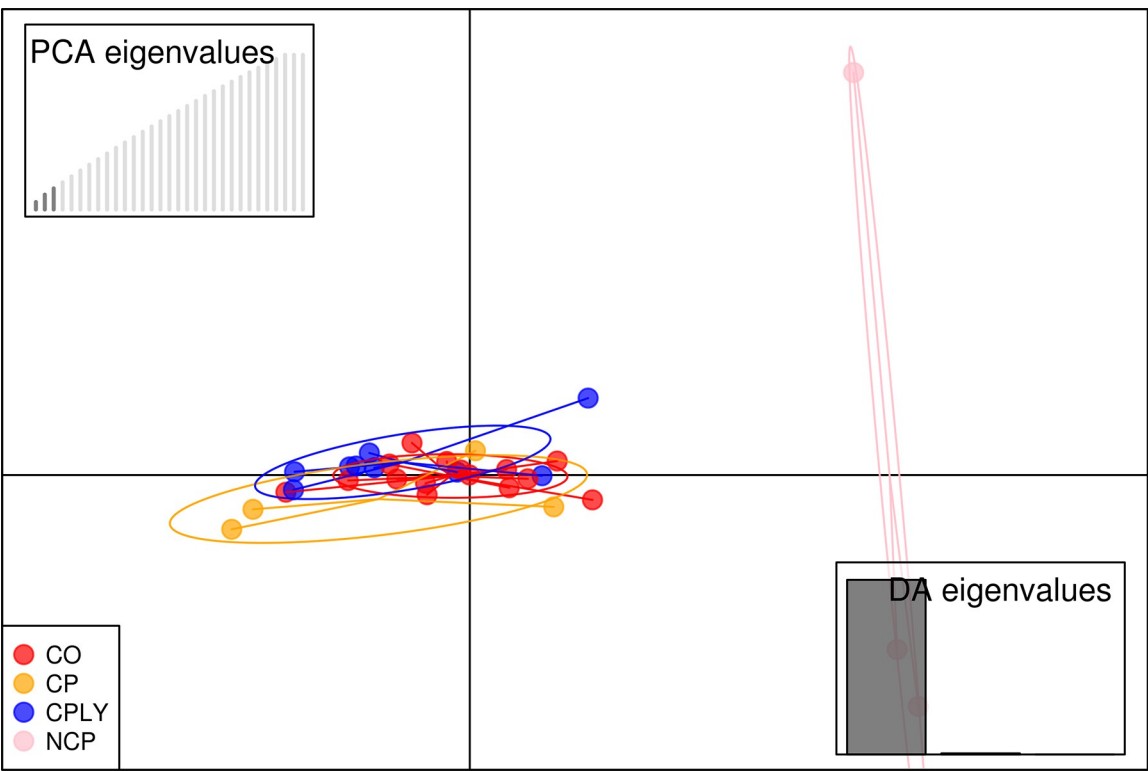

**Fig 3. DAPC scatter plot of investigated brood stocks of *S. caspius*.**

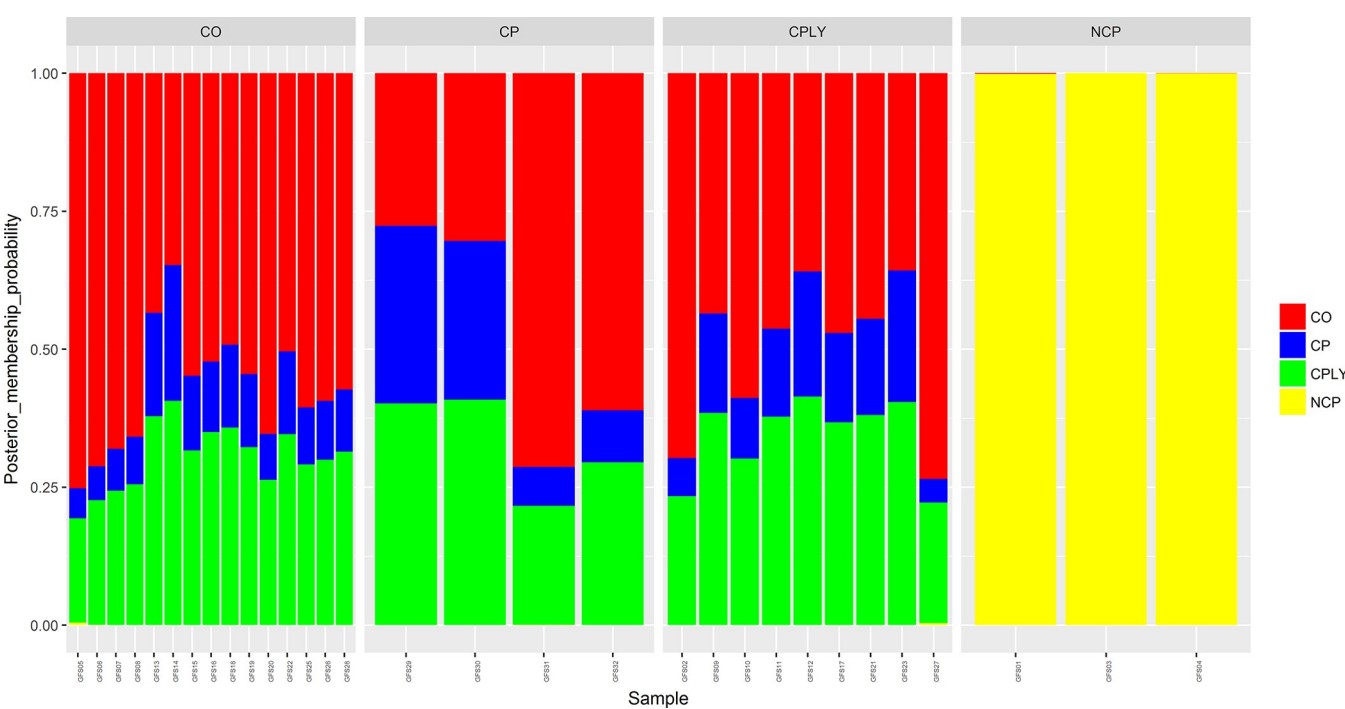

**Fig 4. The admixture plot (K = 2) showing the proportion of genome shared between different clusters.** The X-axis shows all the fish individuals in this study and the Y-axis represents the proportions of the individuals' genome belonging to different ancestral populations.

**Table 1. Genomic information obtained on the brood stocks of *S. caspius*.**

|  | NCP | CPLY | CO | CP |
|---|---|---|---|---|
| Number of Private alleles | 823 | 2107 | 1809 | 1278 |
| Observed heterozygosity | 0.30 | 0.13 | 0.12 | 0.19 |
| Observed homozygosity | 0.7 | 0.87 | 0.88 | 0.81 |
| Expected heterozygosity | 0.34 | 0.36 | 0.37 | 0.33 |
| Expected homozygosity | 0.66 | 0.64 | 0.63 | 0.67 |
| Nucleotide diversity | 0.43 | 0.39 | 0.39 | 0.38 |
| Inbreeding coefficient | 0.24 | 0.65 | 0.72 | 0.41 |

## Discussion

In the present study, the genomic structure of *Salmo caspius* using 62621 SNP markers was investigated at the Kelardasht breeding center in Iran. To the best of our knowledge, this is the first study in the southern basin of the Caspian Sea which aimed to reveal the efficiency of the ongoing restocking activity of *S. caspius* with a high number of genetic markers. The obtained information based on the SNP markers suggests an especial genetic management of the brood stocks of *S. caspius* in the Iranian Restocking center of *S. caspius*.

### Population structure and genetic diversity indices

A high level of admixture pattern was observed in the studied candidates of *S. caspius*. Based on the structural analysis, all investigated fish individuals of *S. caspius* in this study were classifid in two genetically distict populations, so that the NCP group (New candidate parents) could be considered in one clade and the other fish candidates can be placed into the second branch. The observed dispersal of NCP individuals can be caused by the small sample size in this study or the higher genetic differentiation in these individuals, however fish individuals of NCP were placed into one group based on their recent shared genome. The Fst as an index of populations' differentiation ranged from 0.04 to 0.16. According to Wright [37], the Fst value between 0.0–0.05 is an indication for the low differentiation, the Fst between 0.05–0.15 indicates a moderate divergence while the Fst above 0.15 implies a complete separation of the populations. Based on the obtained values of Fst index in this study, populations are in medium level of differentiation with the mean Fst value of 0.08. Having said that, the fish candidates from the NCP group are completely separated from the other candidates with Fst higher than 0.15 which is in line with the high level of genetic differentiation observed in the wild stocks of *S. caspius* in the Aralo-Caspian region [5]. These show that the wild source of *S. caspius* can impose a satisfactory level of genetic diversity but the number of brood stocks in each breeding activity should increase to boast the effective population size and consequent genetic diversity in the next generation.

Considering the genetic diversity, all investigated populations showed a low level of genetic diversity based on the observed heterygosity wherase the NCP group showed the highest

**Table 2. Pair-wise Fst (upper diagonal) and AFD (lower diagonal) indices between different brood stocks of *S. caspius*.**

|  | NCP | CPLY | CO | CP |
|---|---|---|---|---|
| NCP | - | 0.096185 | 0.063366 | 0.163711 |
| CPLY | 0.15625 | - | 0.044616 | 0.086125 |
| CO | 0.03125 | 0.03125 | - | 0.060477 |
| CP | 0.1875 | 0.15625 | 0.125 | - |

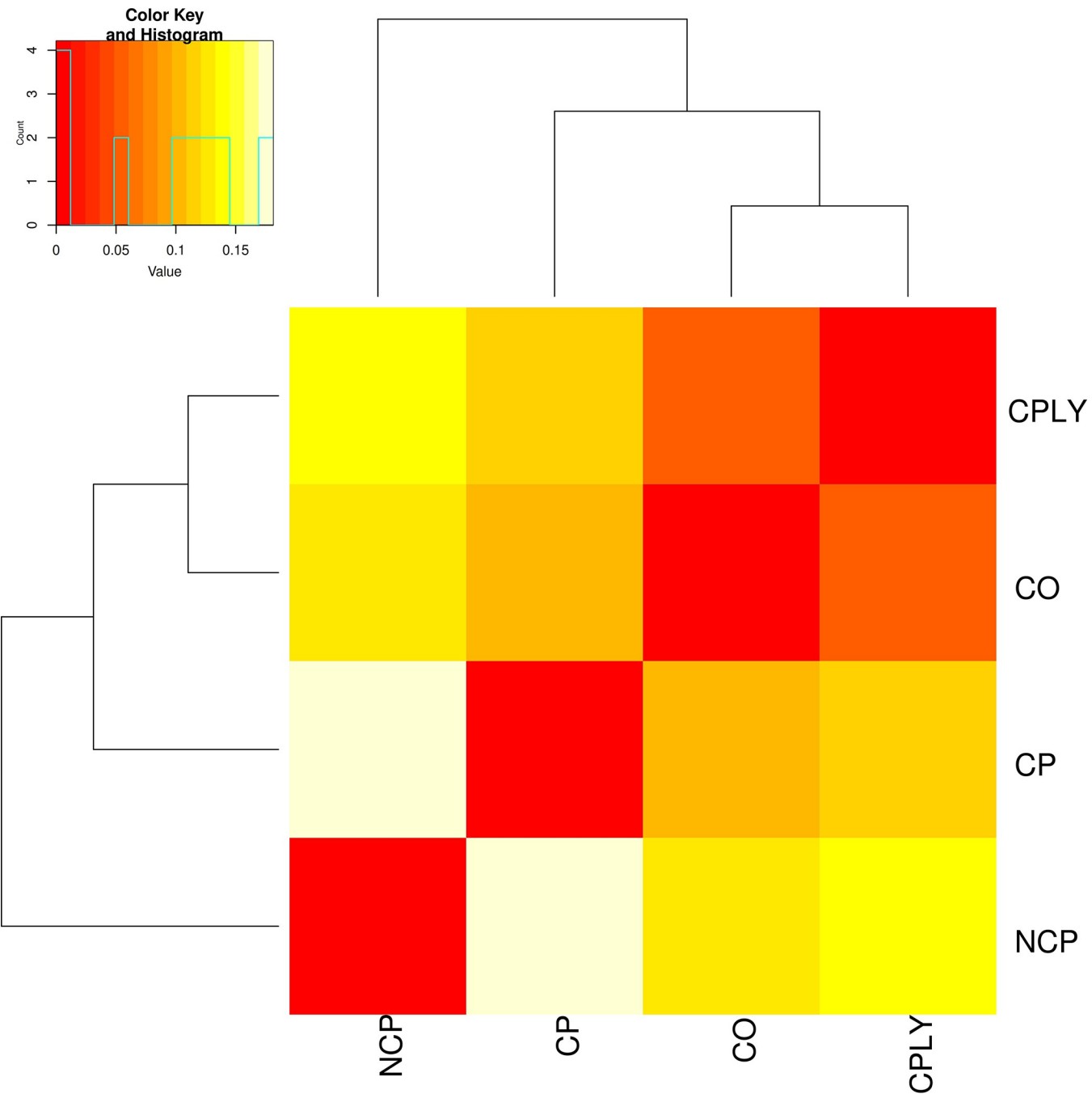

**Fig 5. The UPGMA phylogenetic tree using the nei-based genetic distances between different brood stocks of *s. caspius*.**

heterozygosity. Previously conducted studies on the wild Caspian trout identified different populations of *S. caspius* in two different rivers however they reported a realtive low nucleotide diversity based on D-loop region of mtDNA [38]. In the study on seven different wild and hatchery-based populations of *S. caspius* a satisfactory level of genetic diversity was reported based on the mtDNA control region and nuclear markers (ITS-1 and 10 microsatelite loci), however, it was clearly reported that the current restocking program has not produced

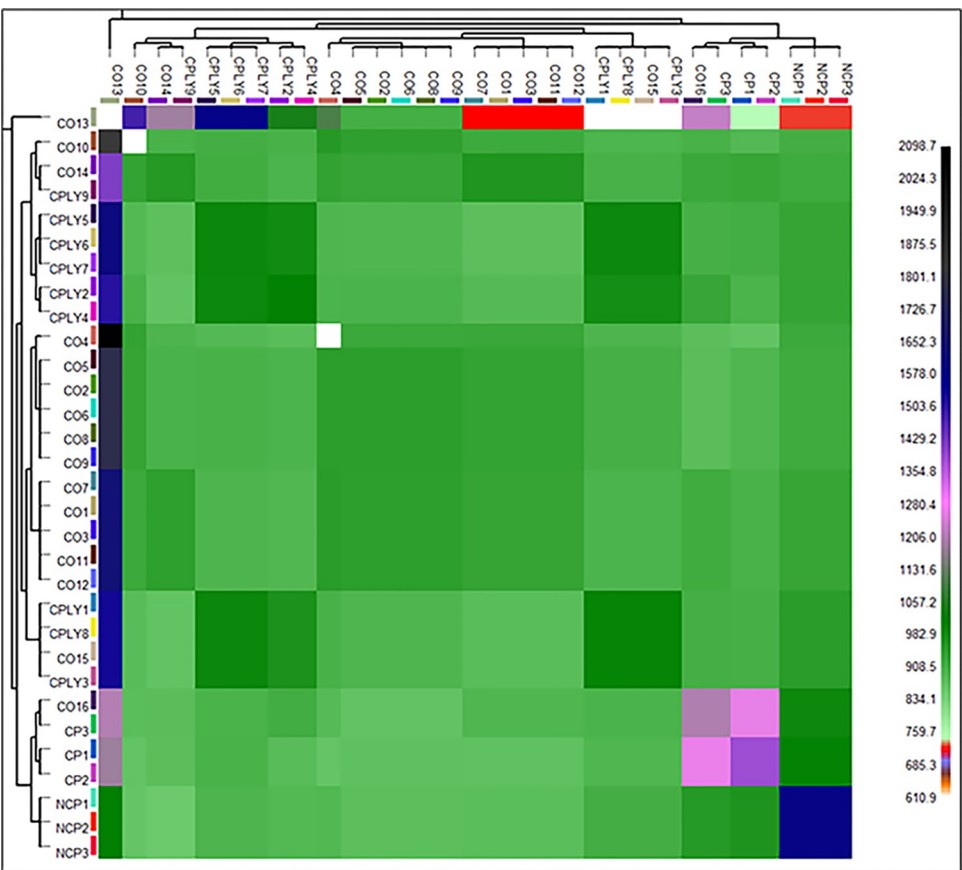

**Fig 6. FineSTRUCTURE classification based on the obtained SNP markers on *S. caspius* in the Kelardasht breeding Center.** Individuals classified along the heatmap's diagonal have common shared coancestry histories and pairwise comparisons outside the diagonal illustrates level of shared coancestry between groups of *S. caspius*.

detectable genetic changes in the wild populations [39]. The results obtained through the present research indicate that the ongoing process of Caspian trout restocking has produced low to medium level of genetic diversity, suggesting an especial attention for expanding the effective population size during the artificial breeding programs to replenish the genetic diversity of *S. caspius* in the southern basin of the Caspian sea.

The current restocking programe of *S. caspius* is mainly based on the maiting of the wild captured parrents in different years (usually two years in a row), then producing brood stock candidates from this mating and in the end using theses produced F0 candidates for breeding schemes with the wild counterparts. Based on the obtained results in this study it seems that the genetic diversity during the undergoing restocking activity of *S. caspius* decreses only after one generation. As it is observed in the CO group, the level of genetic diversity of the candidates offsprings is three times smaller than the new candidate parents from the wild source. On the other hand, positive values for the inbreeding coeficient was observed in the all studied groups, indicating small effective population size both in wild and hatchery-generated populations which results inbreeding depression. It is worthy to note that the breeding scheme of *S. caspius* in term of sex ratio is preferably 1male:3female, however it is totally dependent on the number of available mature adults and the situation. For instance, field surveys showed that during the last year only four migrant female was captured. In spite of the fact that *S. caspius* has faced a drastic reduction in its wild source, still no any information is available on this

species in the IUCN red list. Nevertheless, the Caspian brown trout has been considered as an endangered species in the national level [10,40,41]. Additionally, in accordance with the obtained genetic results in this study it seems that there is a great need to review the restocking activities in order to heal up the wild populations of this species. Restocking is not solely a program based on artificial breeding, and other considerations such as habitat restoration, recovering natural nursury grounds and providing the connectivity among populations across Caspian river basins to enhance the wild populations should be taken into consideration [39]. This integrated conservation strategy is needed as long as the combination of stocking and destruction of trout habitats probably have led to a loss of genetically distinct populations and still is ongoing. Loss of genetyic diversity can diminish the adaptive potential of *S. caspius* which severly threaten its future existence in the Caspian Sea and its related rivers [42].

## Conclusion

The genomic population structure of *Salmo caspius* brood stock in the southern Caspian Sea breeding ceneter was investigated for the first time using 62621 GBS-obtained SNP markers. The obtained results through this study highlighted that current breeding strategy has maintained the low to mdeium level of genetic diversity for *S. caspius* in the southern basin of the Caspian Sea. Positive values were observed for the inbreeding coeficients in all brood stocks is an implication for the small effective population size and a consequence of artificial breeding using realtives.

## Supporting information

**S1 Table. Proportion of missing data for each individual of *S. caspius* in the current study.** (TXT)

## Author Contributions

**Conceptualization:** Mehrshad Zeinalabedini, Omid Jafari.

**Data curation:** Mehrshad Zeinalabedini, Babak Arefnezhad, Maryam Nasrolahpourmoghadam.

**Formal analysis:** Mehrshad Zeinalabedini, Omid Jafari.

**Investigation:** Mehrshad Zeinalabedini.

**Methodology:** Mehrshad Zeinalabedini, Maryam Nafari, Maryam Nasrolahpourmoghadam, Omid Jafari.

**Project administration:** Mehrshad Zeinalabedini.

**Software:** Babak Arefnezhad, Maryam Nafari, Omid Jafari.

**Supervision:** Mehrshad Zeinalabedini.

**Validation:** Mohsen Mardi.

**Visualization:** Babak Arefnezhad, Omid Jafari.

**Writing – original draft:** Arezo Najafikhah.

**Writing – review & editing:** Omid Jafari.

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
