## [Decision Letter · Decision Letter 0]

23 Mar 2023

PONE-D-23-06704GBS-obtained data highlighted the necessity of an updated action plan for wild stocks management of Salmo caspius in the Caspian SeaPLOS ONE

Dear Dr. Jafari,

Thank you for submitting your manuscript to PLOS ONE. After careful consideration, we feel that it has merit but does not fully meet PLOS ONE’s publication criteria as it currently stands. Therefore, we invite you to submit a revised version of the manuscript that addresses the points raised during the review process.

ACADEMIC EDITOR:

Reviewer 1

Dear Authors

I had a chance to act as a reviewer of the ms PONE-D-23-06704 entitled: GBS-obtained data highlighted the necessity of an updated action plan for wild stocks management of Salmo caspius in the Caspian Sea submitted to PlosOne.

This is a study on the genetic diversity of Salmo caspius from a breeding center in the southern Caspian Sea basin. The authors have analysed four different groups of individuals based on the Genotyping-by sequencing (GBS) method. The results are new and interesting. Its publication in Plos One is highly recommended, however, after a significant revision of the present manuscript.

I have made most of the comments and edits in the attached pdf file, but a few main issues are listed below:

1- There are too many knowledge gaps in the introduction section (see the comments), which need further clarifications.

2- The sampling section needs improvement. As a reader, I have no idea what the four study groups are (NCP, NCPLY, CO, and CP). The authors need to explain these groups in more detail, and also explain the rationale for this grouping. This part needs to be explained carefully and comprehensively, since the whole study seems to be based on it.

3- Finally, there is a bias in the sampling. The CO and NCP groups are represented by 4 and 3 individuals respectively, whereas this is contrasted by the fact that in a standard population genetic analysis each population/group must be represented by at least 10 individuals. If possible, the number of analyzed samples should be increased for these two groups.

Sincerely Yours

6. PLOS authors have the option to publish the peer review history of their article (what does this mean?). If published, this will include your full peer review and any attached files.

Do you want your identity to be public for this peer review? For information about this choice, including consent withdrawal, please see our Privacy Policy. No

Confidential to Editor

1. Do you have any potential or perceived competing interests that may influence your review? Please review our Competing Interests policy and declare any potential interests that you feel the Editor should be aware of when considering your review. If you have no competing interests, please write "I have no competing interests." Dear Editor

PlosOne

Thanks for considering me as a reviewer of the ms PONE-D-23-06704 entitled: GBS-obtained data highlighted the necessity of an updated action plan for wild stocks management of Salmo caspius in the Caspian Sea submitted to PlosOne.

This is a study on the genetic diversity of Salmo caspius from a breeding center in the southern Caspian Sea basin. The authors have analysed four different groups of individuals based on the Genotyping-by sequencing (GBS) method. The results are new and interesting. Its publication in Plos One is highly recommended, however, after a significant revision of the present manuscript.

I have made most of the comments and edits in the attached pdf file, but a few main issues are listed below:

1- There are too many knowledge gaps in the introduction section (see the comments), which need further clarifications.

2- The sampling section needs improvement. As a reader, I have no idea what the four study groups are (NCP, NCPLY, CO, and CP). The authors need to explain these groups in more detail, and also explain the rationale for this grouping. This part needs to be explained carefully and comprehensively, since the whole study seems to be based on it.

3- Finally, there is a bias in the sampling. The CO and NCP groups are represented by 4 and 3 individuals respectively, whereas this is contrasted by the fact that in a standard population genetic analysis each population/group must be represented by at least 10 individuals. If possible, the number of analyzed samples should be increased for these two groups.

Reviewer 2

The manuscript by Jafari et al. deals with population genetic analysis of a few groups from a hatchery in the south Caspian Sea basin that can provide data for stock augmentation purposes. Based on a fast review of the manuscript, I found major problems with taxonomic aspects, data analyses and also the claims made in the manuscript. I could not see and statistical support for the claims of the authors. The most important analysis for a conservation oriented work is the analysis of molecular variance (AMOVA) which is lacking. Or the genetic distance that is used has is not a good choice due to small sample sizes used in this study. Apparently, authors were not familiar with genomic works performed on the Caspian Sea trout in recent years and ignored them completely. Further, the manuscript needs language check by a native English speaking person.

Overall, I cannot accept the manuscript for publication in its current form. I suggest major revisions and resubmission.

My comments are listed bellow:

The title should be revised, may be: Genome-wide data suggest a revision in management of the Caspian Sea trout Salmo caspius

Abstract

Line 29: provide also authorship at first mention of the scientific names!

Line 34: GBS-obtained SNP variants: replace with genome-wide single nucleotide polymorphisms (SNP)

Line 36: admixture

38: critically endangered? It is not even listed in IUCN redlist. Please check it!

Introduction

Line 43: families

Line 45: consequence

Lines 45-47: no! Salmo trutta is the Atlantic trout! All natural populations of brown trout in the southern Caspian Sea basin are Salmo caspius: resident and sea-run forms are only ecological forms, but not species! Please rather than using general checklists, have a look at recent papers published on population genomics of Salmo caspius using the same approach used here:

Hashemzadeh Segherloo, I., Tabatabaei, S.N., Abdoli, A. et al. Biogeographic insights from a genomic survey of Salmo trouts from the Aralo-Caspian regions. Hydrobiologia 849, 4325–4339 (2022). https://doi.org/10.1007/s10750-022-04993-8

Hashemzadeh Segherloo, I., Freyhof, J., Berrebi, P., Ferchaud, A.L., Geiger, M., Laroche, J., Levin, B.A., Normandeau, E. and Bernatchez, L., 2021. A genomic perspective on an old question: Salmo trouts or Salmo trutta (Teleostei: Salmonidae)?. Molecular Phylogenetics and Evolution, 162, p.107204.

Tabatabaei, S.N., Abdoli, A., Hashemzadeh Segherloo, I., Normandeau, E., Ahmadzadeh, F., Nejat, F. and Bernatchez, L., 2020. Fine-scale population genetic structure of Endangered Caspian Sea trout, Salmo caspius: implications for conservation. Hydrobiologia, 847, pp.3339-3353.

Tabatabaei, S.N., Abdoli, A., Ahmadzadeh, F., Primmer, C.R., Swatdipong, A. and Segherloo, I.H., 2020. Mixed stock assessment of lake-run Caspian Sea trout Salmo caspius in the Lar National Park, Iran. Fisheries Research, 230, p.105644.

Lines 47-49: I could not see any genetic work in references, which are cited (checklists). Please use references dealing directly with the Caspian sea trout using robust data. It is clear that resident form of the Caspian trout or any other member of Salmo trutta species-complex is morphologically different from lake- or sea-run forms. Please see the above mentioned references.

Lines 62-65: I guess these are not morphotypes, are they? They should be temporal ecotypes, which migrate during different seasons. Please clarify!

Line 76: please provide the complete terms at the first mention and then use only abbreviations: Next Generation Sequencing (NGS)

Lines 76-77: please provide references.

Line 79: only two citations show the wide use of the approach in population genetics and aquaculture? Please provide examples!

Line 84: breeding center or centers across the southern Caspian Sea basin? If more than one breeding center is considered, please indicate. In m&m section authors indicate only one breeding center!

Material and Methods

Lines 87-91: it is not clear how these fish were selected? where the fish had been caught from? Were they all from the same population? What was the relationship of different groups? What does it mean “new candidate parents”, “candidate parents last year”, “candidate offspring”, and “captured parents”? please clarify! How many specimens from each group? …

Line 97: not guideline, but protocol or manual

Line 98: please provide webpage of the manufacturer.

Line 102: please refer to citations properly! Jafari et al.

Line 103: webpage for BGI co.

Line 114: hired? Used?

Lines 135-138: as sample sizes used in this study are small, it is suggested to use other metrics like allele frequency differences to avoid problems related to small sample size. Fst is sensitive to small sample sizes.

Line 139: taken or calculated?

Results:

There are no details on proportion of missing data and how it was managed! Please clarify!

Lines 142-148: which K was selected in Structure analysis? Please clarify and provide statistical evidence.

The work aims at conservation, but one of the most important statistical analyses which is mostly used in conservation genetics works is missing: I expected to see AMOVA analysis of the data to see statistical support for population structure …

Line 179: Fig. 4a? appears to be 5a

Discussion

Lines 186-189: please see my previous comments: it is not the first genomic work on the Caspian Sea trout. There had been other studies!

Line 194: structure analysis

Lines 196-199: Fst is prone to small sample sizes, see previous comments!

Line 218: based

220-224: What is the mating scheme in the noted hatchery? The reduced diversity may be a result of breeding scheme! What was the sex ratio? How many brood fish had been used? And so on. All these are important factors which must be clarified.

Lines 226-227: It is not listed in the redlist as I searched! Please check!

Data availability

Line 248: data should be deposited in public databases like SRA or VCF files can be uploaded to Dryad …

Ethics:

Lines 251-252: Was there a licence issued by the Ethics committee? Pleas provide

References

References do not follow a standard and unified format. In some cases they are incomplete and details are missing!

Figure 1 why members of the same cluster (NCP) are distributed all around the graph? Is there any justification?

Journal Requirements: 

2. To comply with PLOS ONE submissions requirements, in your Methods section, please provide additional information regarding the experiments involving animals and ensure you have included details on (1) methods of sacrifice, (2) methods of anesthesia and/or analgesia, and (3) efforts to alleviate suffering

4. We note that Figure (1) in your submission contain copyrighted images. All PLOS content is published under the Creative Commons Attribution License (CC BY 4.0), which means that the manuscript, images, and Supporting Information files will be freely available online, and any third party is permitted to access, download, copy, distribute, and use these materials in any way, even commercially, with proper attribution. For more information, see our copyright guidelines: http://journals.plos.org/plosone/s/licenses-and-copyright.

1. You may seek permission from the original copyright holder of Figure(s) [#] to publish the content specifically under the CC BY 4.0 license. 

"No"

 This information should be included in your cover letter; we will change the online submission form on your behalf

Reviewers' comments:

Reviewer's Responses to Questions

**Comments to the Author**

1. Is the manuscript technically sound, and do the data support the conclusions?

Reviewer #1: Yes

Reviewer #2: Partly

2. Has the statistical analysis been performed appropriately and rigorously?

Reviewer #1: Yes

Reviewer #2: No

3. Have the authors made all data underlying the findings in their manuscript fully available?

Reviewer #1: Yes

Reviewer #2: Yes

4. Is the manuscript presented in an intelligible fashion and written in standard English?

Reviewer #1: No

Reviewer #2: No

5. Review Comments to the Author

Reviewer #1: Dear Authors

I had a chance to act as a reviewer of the ms PONE-D-23-06704 entitled: GBS-obtained data highlighted the necessity of an updated action plan for wild stocks management of Salmo caspius in the Caspian Sea submitted to PlosOne.

This is a study on the genetic diversity of Salmo caspius from a breeding center in the southern Caspian Sea basin. The authors have analysed four different groups of individuals based on the Genotyping-by sequencing (GBS) method. The results are new and interesting. Its publication in Plos One is highly recommended, however, after a significant revision of the present manuscript.

I have made most of the comments and edits in the attached pdf file, but a few main issues are listed below:

1- There are too many knowledge gaps in the introduction section (see the comments), which need further clarifications.

2- The sampling section needs improvement. As a reader, I have no idea what the four study groups are (NCP, NCPLY, CO, and CP). The authors need to explain these groups in more detail, and also explain the rationale for this grouping. This part needs to be explained carefully and comprehensively, since the whole study seems to be based on it.

3- Finally, there is a bias in the sampling. The CO and NCP groups are represented by 4 and 3 individuals respectively, whereas this is contrasted by the fact that in a standard population genetic analysis each population/group must be represented by at least 10 individuals. If possible, the number of analyzed samples should be increased for these two groups.

Sincerely Yours

Reviewer #2: The manuscript by Jafari et al. deals with population genetic analysis of a few groups from a hatchery in the south Caspian Sea basin that can provide data for stock augmentation purposes. Based on a fast review of the manuscript, I found major problems with taxonomic aspects, data analyses and also the claims made in the manuscript. I could not see and statistical support for the claims of the authors. The most important analysis for a conservation oriented work is the analysis of molecular variance (AMOVA) which is lacking. Or the genetic distance that is used has is not a good choice due to small sample sizes used in this study. Apparently, authors were not familiar with genomic works performed on the Caspian Sea trout in recent years and ignored them completely. Further, the manuscript needs language check by a native English speaking person.

Overall, I cannot accept the manuscript for publication in its current form. I suggest major revisions and resubmission.

My comments are listed bellow:

The title should be revised, may be: Genome-wide data suggest a revision in management of the Caspian Sea trout Salmo caspius

Abstract

Line 29: provide also authorship at first mention of the scientific names!

Line 34: GBS-obtained SNP variants: replace with genome-wide single nucleotide polymorphisms (SNP)

Line 36: admixture

38: critically endangered? It is not even listed in IUCN redlist. Please check it!

Introduction

Line 43: families

Line 45: consequence

Lines 45-47: no! Salmo trutta is the Atlantic trout! All natural populations of brown trout in the southern Caspian Sea basin are Salmo caspius: resident and sea-run forms are only ecological forms, but not species! Please rather than using general checklists, have a look at recent papers published on population genomics of Salmo caspius using the same approach used here:

Hashemzadeh Segherloo, I., Tabatabaei, S.N., Abdoli, A. et al. Biogeographic insights from a genomic survey of Salmo trouts from the Aralo-Caspian regions. Hydrobiologia 849, 4325–4339 (2022). https://doi.org/10.1007/s10750-022-04993-8

Hashemzadeh Segherloo, I., Freyhof, J., Berrebi, P., Ferchaud, A.L., Geiger, M., Laroche, J., Levin, B.A., Normandeau, E. and Bernatchez, L., 2021. A genomic perspective on an old question: Salmo trouts or Salmo trutta (Teleostei: Salmonidae)?. Molecular Phylogenetics and Evolution, 162, p.107204.

Tabatabaei, S.N., Abdoli, A., Hashemzadeh Segherloo, I., Normandeau, E., Ahmadzadeh, F., Nejat, F. and Bernatchez, L., 2020. Fine-scale population genetic structure of Endangered Caspian Sea trout, Salmo caspius: implications for conservation. Hydrobiologia, 847, pp.3339-3353.

Tabatabaei, S.N., Abdoli, A., Ahmadzadeh, F., Primmer, C.R., Swatdipong, A. and Segherloo, I.H., 2020. Mixed stock assessment of lake-run Caspian Sea trout Salmo caspius in the Lar National Park, Iran. Fisheries Research, 230, p.105644.

Lines 47-49: I could not see any genetic work in references, which are cited (checklists). Please use references dealing directly with the Caspian sea trout using robust data. It is clear that resident form of the Caspian trout or any other member of Salmo trutta species-complex is morphologically different from lake- or sea-run forms. Please see the above mentioned references.

Lines 62-65: I guess these are not morphotypes, are they? They should be temporal ecotypes, which migrate during different seasons. Please clarify!

Line 76: please provide the complete terms at the first mention and then use only abbreviations: Next Generation Sequencing (NGS)

Lines 76-77: please provide references.

Line 79: only two citations show the wide use of the approach in population genetics and aquaculture? Please provide examples!

Line 84: breeding center or centers across the southern Caspian Sea basin? If more than one breeding center is considered, please indicate. In m&m section authors indicate only one breeding center!

Material and Methods

Lines 87-91: it is not clear how these fish were selected? where the fish had been caught from? Were they all from the same population? What was the relationship of different groups? What does it mean “new candidate parents”, “candidate parents last year”, “candidate offspring”, and “captured parents”? please clarify! How many specimens from each group? …

Line 97: not guideline, but protocol or manual

Line 98: please provide webpage of the manufacturer.

Line 102: please refer to citations properly! Jafari et al.

Line 103: webpage for BGI co.

Line 114: hired? Used?

Lines 135-138: as sample sizes used in this study are small, it is suggested to use other metrics like allele frequency differences to avoid problems related to small sample size. Fst is sensitive to small sample sizes.

Line 139: taken or calculated?

Results:

There are no details on proportion of missing data and how it was managed! Please clarify!

Lines 142-148: which K was selected in Structure analysis? Please clarify and provide statistical evidence.

The work aims at conservation, but one of the most important statistical analyses which is mostly used in conservation genetics works is missing: I expected to see AMOVA analysis of the data to see statistical support for population structure …

Line 179: Fig. 4a? appears to be 5a

Discussion

Lines 186-189: please see my previous comments: it is not the first genomic work on the Caspian Sea trout. There had been other studies!

Line 194: structure analysis

Lines 196-199: Fst is prone to small sample sizes, see previous comments!

Line 218: based

220-224: What is the mating scheme in the noted hatchery? The reduced diversity may be a result of breeding scheme! What was the sex ratio? How many brood fish had been used? And so on. All these are important factors which must be clarified.

Lines 226-227: It is not listed in the redlist as I searched! Please check!

Data availability

Line 248: data should be deposited in public databases like SRA or VCF files can be uploaded to Dryad …

Ethics:

Lines 251-252: Was there a licence issued by the Ethics committee? Pleas provide

References

References do not follow a standard and unified format. In some cases they are incomplete and details are missing!

Figure 1 why members of the same cluster (NCP) are distributed all around the graph? Is there any justification?

We look forward to receiving your revised manuscript.

Kind regards,

Ishtiyaq Ahmad, Ph.D

Academic Editor

PLOS ONE

Journal Requirements:

Additional Editor Comments (if provided):

6. PLOS authors have the option to publish the peer review history of their article (what does this mean?). If published, this will include your full peer review and any attached files.

Reviewer #1: No

Reviewer #2: No

---

## [Author Response · Author response to Decision Letter 0]

30 May 2023

Before everything, authors are completely grateful for the peer-review of the manuscript. All comments have been seriously taken into consideration and are addressed in the manuscript.

Reviewer 1:

1- There are too many knowledge gaps in the introduction section (see the comments), which need further clarifications.

Response: We have made several changes and added new information into the introduction section to avoid the problem of knowledge gap.

2- The sampling section needs improvement. As a reader, I have no idea what the four study groups are (NCP, NCPLY, CO, and CP). The authors need to explain these groups in more detail, and also explain the rationale for this grouping. This part needs to be explained carefully and comprehensively, since the whole study seems to be based on it.

Response: The sampling section has been improved, specifically with the all details on the way of grouping.

3- Finally, there is a bias in the sampling. The CO and NCP groups are represented by 4 and 3 individuals respectively, whereas this is contrasted by the fact that in a standard population genetic analysis each population/group must be represented by at least 10 individuals. If possible, the number of analyzed samples should be increased for these two groups.

Response: Thank you for all your helpful comments. Unfortunately, it was not possible for us during the sampling to have more specimens because of several issues. Of course, availability of more samples can enrich the work but for example, in the year of sampling only 3 mature adults had been caught which disabled us to have more.

Reviewer 2:

1. The title should be revised, may be: Genome-wide data suggest a revision in management of the Caspian Sea trout Salmo caspius.

Response: The title has been changed to “Genome-wide data suggest a revision in management of the Caspian Sea trout Salmo caspius”

Abstract

Line 29: provide also authorship at first mention of the scientific names!

The authorship was provided: Salmo caspius Kessler, 1877

Line 34: GBS-obtained SNP variants: replace with genome-wide single nucleotide polymorphisms (SNP)

Response: Done.

Line 36: admixture

Response: Done.

Line 38: critically endangered? It is not even listed in IUCN redlist. Please check it!

Response: It has been modified to the nationally endangered species.

Introduction

Line 43: families

Response: Done.

Line 45: consequence

Response: Done.

Lines 45-47: no! Salmo trutta is the Atlantic trout! All natural populations of brown trout in the southern Caspian Sea basin are Salmo caspius: resident and sea-run forms are only ecological forms, but not species! Please rather than using general checklists, have a look at recent papers published on population genomics of Salmo caspius using the same approach used here.

Response: Thank you so much for the great comment. We have been referring to the published check list but now we changed to whole introduction based on the recent published papers on salmo caspius using GBS data.

Lines 47-49: I could not see any genetic work in references, which are cited (checklists). Please use references dealing directly with the Caspian Sea trout using robust data. It is clear that resident form of the Caspian trout or any other member of Salmo trutta species-complex is morphologically different from lake- or sea-run forms. Please see the above-mentioned references.

Response: The introduction section has been modified as you nicely suggested based on the provided references and all of them are cited now in the manuscript. Their reference numbers are: 5, 8, 9 and 11.

Lines 62-65: I guess these are not morphotypes, are they? They should be temporal ecotypes, which migrate during different seasons. Please clarify!

Response: It seems to us that the migratory behavior can be considered as a trait/morphotype, and that is why we mentioned two morphotypes. Now, it is changed to ecological forms in line 54.

Line 76: please provide the complete terms at the first mention and then use only abbreviations: Next Generation Sequencing (NGS)

Response: Done.

Lines 76-77: please provide references.

Response: Two references (17 and 18) have been mentioned in line 87.

Line 79: only two citations show the wide use of the approach in population genetics and aquaculture? Please provide examples!

Response: Some examples concerning the usage of GBS data in Fisheries sciences have been provided in lines 89-92.

Line 84: breeding center or centers across the southern Caspian Sea basin? If more than one breeding center is considered, please indicate. In m&m section authors indicate only one breeding center!

Response: Only one breeding center was considered. Indeed, Kelardasht is the one and only center for restocking Salmo caspius in the southern Caspian Sea.

Material and Methods

Lines 87-91: it is not clear how these fish were selected? where the fish had been caught from? Were they all from the same population? What was the relationship of different groups? What does it mean “new candidate parents”, “candidate parents last year”, “candidate offspring”, and “captured parents”? please clarify! How many specimens from each group? 

Response: The new explanation for material and methods section has been provided in lines 100-106.

Line 97: not guideline, but protocol or manual

Response: Done (in line 114).

Line 98: please provide webpage of the manufacturer.

Response: Done (in lines 114 and 116).

Response: 

Line 102: please refer to citations properly! Jafari et al.

Response: Done.

Line 103: webpage for BGI co.

Response: The related web page is available in line 121.

Line 114: hired? Used?

Response: “hired” was changed to “used” in line 132.

Lines 135-138: as sample sizes used in this study are small, it is suggested to use other metrics like allele frequency differences to avoid problems related to small sample size. Fst is sensitive to small sample sizes.

Response: We appreciate your very informative comments. The AFD (Allele Frequency Differences) between pairs of groups was estimated and provided now in the manuscript both in m&m section and the results (157-158, and 188-189, and Table 2).

Line 139: taken or calculated?

Response: "taken" was changed to "calculated".

Results:

There are no details on proportion of missing data and how it was managed! Please clarify!

Response: The proportion of missing data has been provided in supplementary information as named S12_Table. Also, during the Bioinformatic analysis, the way in which missing values were behaved has been explained in lines 160-161.

Lines 142-148: which K was selected in Structure analysis? Please clarify and provide statistical evidence.

Response: K= 2 was considered for the structure analysis based on the obtained lowest BIC. The related explanations are available in lines 151-152 and 177.

The work aims at conservation, but one of the most important statistical analyses which is mostly used in conservation genetics works is missing: I expected to see AMOVA analysis of the data to see statistical support for population structure.

Response: The AMOVA analysis has been added to the manuscript which was done using Genodive. The related sentences are in lines 158-161 and 189-190.

Line 179: Fig. 4a? appears to be 5a

Response: Done.

Discussion

Lines 186-189: please see my previous comments: it is not the first genomic work on the Caspian Sea trout. There had been other studies!

Response: Here our main focus was on the current restocking activity which is held in the Kelardasht restocking activity. So, we changed it to " To the best of our knowledge, this is the first study in the southern basin of the Caspian Sea which aimed to reveal the efficiency of the ongoing restocking activity of S. caspius with a high number of genetic markers."

Line 194: structure analysis

Response: Done.

Lines 196-199: Fst is prone to small sample sizes, see previous comments!

Response: As you nicely suggested, the AFD analysis has been added to the manuscript.

Line 218: based

Response: Done.

220-224: What is the mating scheme in the noted hatchery? The reduced diversity may be a result of breeding scheme! What was the sex ratio? How many brood fish had been used? And so on. All these are important factors which must be clarified.

Response: The related explanations are available in lines 254- 260.

Lines 226-227: It is not listed in the redlist as I searched! Please check!

Response: In lines 258-260 the revised sentences are provided (In spite of the fact that S. caspius has faced a drastic reduction in its wild source, still no any information is available on this species in the IUCN red list. Nevertheless, the Caspian brown trout has been considered as an endangered species in the national level).

Data availability

Line 248: data should be deposited in public databases like SRA or VCF files can be uploaded to Dryad …

Response: The raw data are deposited in SRA and now is indicated in the manuscript (PRJNA966795).

Ethics:

Lines 251-252: Was there a licence issued by the Ethics committee? Pleas provide

References

Response: The animal study was reviewed and approved by the Ethics Committee of Agricultural Biotechology Reasearch Institute of Iran (NO. 014-05-05-006-94006).

References do not follow a standard and unified format. In some cases, they are incomplete and details are missing!

Response: Thank you for your careful attention. We have used the endnote style of the Journal to avoid unwanted typo mistakes. 

Figure 1 why members of the same cluster (NCP) are distributed all around the graph? Is there any justification?

Response: Concerning the high dispersal of NCP group in PCA scatter plot, it can be linked to the higher genetic diversity by NCP group, however, all other structural analysis such as fineRADstructure based on their recent shared genome put them in the same clade.

---

## [Decision Letter · Decision Letter 1]

13 Jun 2023

Genome-wide data suggest a revision in management of the Caspian Sea trout Salmo caspius

PONE-D-23-06704R1

Dear Dr. Jafri,

We’re pleased to inform you that your manuscript has been judged scientifically suitable for publication and will be formally accepted for publication once it meets all outstanding technical requirements.

Kind regards,

Ishtiyaq Ahmad, Ph.D

Academic Editor

PLOS ONE

Additional Editor Comments (optional):

Authors have adreesed all queries as per reviewer suggestions. So, I recommend your paper for publication.

**Comments to the Author**

Reviewer #1: All comments have been addressed

2. Is the manuscript technically sound, and do the data support the conclusions?

Reviewer #1: Yes

3. Has the statistical analysis been performed appropriately and rigorously? 

Reviewer #1: Yes

4. Have the authors made all data underlying the findings in their manuscript fully available?

Reviewer #1: Yes

5. Is the manuscript presented in an intelligible fashion and written in standard English?

Reviewer #1: Yes

6. Review Comments to the Author

Reviewer #1: Dear Authors

I had a chance to review the revised version of the ms PONE-D-23-06704R1, entitled "Genome-wide data suggest a revision in management of the Caspian Sea trout Salmo caspius". It seems that you have implemented the comments/corrections based on the reviewer's comments.

I found it very interesting and now it can be considered for publication.

However, journal formatting should be done.

Best

7. PLOS authors have the option to publish the peer review history of their article (what does this mean?). If published, this will include your full peer review and any attached files.

Reviewer #1: **Yes: **Prof. Hamid Reza Esmaeili

---

## [Editor Report · Acceptance letter]

13 Jul 2023

PONE-D-23-06704R1 

Genome-wide data suggest a revision in management of the Caspian Sea trout *Salmo caspius*

Dear Dr. Jafari:

I'm pleased to inform you that your manuscript has been deemed suitable for publication in PLOS ONE. Congratulations! Your manuscript is now with our production department. 

Kind regards, 

on behalf of

Dr. Ishtiyaq Ahmad 

Academic Editor

PLOS ONE